# Nurse's perceptions of support for sexual and reproductive issues in adolescents and young adults with cancer

Akiko Tomioka[1]*, Kyoko Obama[2], Hiromi Okada[1‡], Eiko Yamauchi[3‡], Kimiko Iwase[4‡], Mitsue Maru[5]

1 Faculty of Healthcare, Tokyo Healthcare University, Tokyo, Japan, 2 Institute for Cancer Control, National Cancer Center, Tokyo, Japan, 3 Faculty of Medicine, Ehime University, Matsuyama, Ehime, Japan, 4 Faculty of Nursing & Rehabilitation, Konan Women's University, Kobe, Hyogo, Japan, 5 College of Nursing Art & Science, University of Hyogo, Kobe, Japan

◔ These authors contributed equally to this work.
‡ HO, EY and KI also contributed equally to this work.
* a-tomioka@thcu.ac.jp

**Data Availability Statement:** All relevant data are within the paper and its Supporting information files.

## Abstract

Adolescent and young adult (AYA) with cancer are at risk for developing sexual and reproductive problems; therefore, they have special needs. AYA with cancer treated in both pediatric and adult wards are a minority in Japan; thus, accumulating experience for supporting this unique patient population is difficult for nurses. Hence, this study aimed to clarify nurses' perceptions on support for sexual and reproductive issues among AYA with cancer. A questionnaire survey was administered to nurses at designated cancer hospitals across Japan who had been working for at least 1 year in a department involved in the treatment or follow-up of patients aged 15–39 years. Nurses were asked regarding their perceptions on support for sexual and reproductive issues faced by AYA with cancer. A total of 865 nurses responded to this survey; nurses affiliated with adult departments, those with more experience in cancer nursing, those affiliated with cancer-related academic and professional societies, and certified nurse specialists or certified nurses significantly recognized insufficient support for sexual and reproductive issues. However, nurses were hesitant and found it difficult to intervene in such issues. Nurses recognized the importance of providing support for sexual and reproductive issues but faced difficulties in addressing them. They need to discuss these issues and improve the care provided to AYA with cancer.

## Introduction

Adolescent and young adult (AYA) experience life events, including going to school or work, marrying, having children, and having a unique set of needs, at this life stage. A diagnosis and subsequent treatment of cancer during this life stage inevitably disrupt their outlook on life. Moreover, cancer treatments, such as chemotherapy and radiation, may cause infertility and sexual dysfunction [1, 2]. Recent developments in assisted reproduction technologies have

**Funding:** This work was supported by the Ministry of Health, Labour and Welfare Sciences Research Grant to Keizo Horibe (H27-Ippan-005) as a part of the grant titled "Research on comprehensive cancer treatment and support for adolescents and young adults (AYA)". The funder had no role in study design, data collection and analysis, decision to publish, or preparation of the manuscript.

**Competing interests:** The authors have declared that no competing interests exist.

been remarkable, and collaboration between cancer and reproductive medicine, such as fertility preservation therapy, is often recommended [3, 4]. Moreover, the need to improve patient decision-making and support systems for fertility preservation has been suggested [5]. Reports have indicated that AYA with cancer are greatly concerned with issues regarding sexual and reproductive function and remain uncertain about their future [6–9]. For AYA, the decline in sexual and reproductive function is a problem that is not only functional in nature but also deeply related to the development of their sexuality and identity [10].

In Japan, the Cancer Control Act and the designation of cancer hospitals was implemented in 2002 to promote standardized, high-quality cancer treatment nationwide. In 2018, the strengthening of the support system for AYA aged 15–39 years was recommended. However, studies have estimated that approximately 3% of the total numbers of patients with cancer in Japan are AYA [11, 12], and only a limited number of inpatient units have been specifically designed for AYA with cancer. Moreover, because a small number of patients are treated in each department, accumulating experience for supporting AYA is difficult for nurses. Hence, the present study aimed to elucidate nurses' perceptions on the support for sexual and reproductive issues in AYA with cancer to help establish measures for improving care.

## Materials and methods

### Participants

Participants were nurses who worked at designated cancer hospitals in Japan and had more than a year's experience in a department that treated or continuously followed up AYA with cancer.

### Procedure

Among the 427 nursing directors from designated cancer hospitals across Japan who were contacted, 65 agreed to participate in this survey. The designated number of surveys was then mailed to these 65 facilities, and the nursing director was tasked with distributing the survey to nurses in their departments. The questionnaires were anonymous and collected by postal mail in sealed envelopes. The survey period was from June to August 2016.

### Measurements

This study was part of a large survey across Japan. The questionnaire had six sections: a) care difficulties for AYA cancer patients/survivors, b) perceived needs of patients/survivors in daily care, c) support for sexual and reproductive issues, d) end-of-life care, e) characteristics of AYA patients/survivors whom nurses felt care difficulties, and f) facilitating/disturbing factors on the quality of care for AYAs. In this study, we used the questionnaire section "support for sexual and reproductive issues."

To investigate the nurses' perception of support for sexual and reproductive issues faced by AYA with cancer, four questions were asked (Q1: "Do you think that adolescent patients with cancer [age, 15–19 years] are sufficiently debriefed with necessary information regarding their sexual and reproductive functions?"; Q2: "Do you think that young adult patients with cancer [age, 20–39 years] are sufficiently debriefed with the necessary information regarding their sexual and reproductive functions?"; Q3: "Do you think that examinations and care are conducted sufficiently taking sexuality into consideration?"; and Q4: "Do you think that a sufficient support system for sexual and reproductive functions is in place?") and scored as follows: "Sufficient" (1 point), "Moderately Sufficient" (2 points), "Moderately Insufficient" (3 points),

and "Insufficient" (4 points). An option for answering "I don't know" was added to determine whether nurses truly understood the existing circumstances.

Participants were asked to freely describe in writing the issues and difficulties they experienced in supporting sexual and reproductive problems faced by AYA with cancer.

To determine the participants' attributes, we asked the following information: department, job title and qualification, years of experience as a nurse, years of experience in cancer nursing, and affiliation with cancer-related academic and professional societies.

The researcher's group consisting of four pediatric cancer nurses, two adult cancer nurses, and two clinical nurses reviewed the questionnaire for surface and content validity.

## Data analysis

Mean scores and standard deviations for each question were calculated. However, the following participants were excluded: those who had less than a year of nursing experience, those who did not answer some questions, and those who answered "I don't know." Thereafter, the total score for each question was calculated, with higher scores indicating a greater insufficiency of support for sexual and reproductive concerns among AYA with cancer. Cronbach's coefficient α was calculated to confirm the internal consistency of each item. Multiple comparisons were performed using Bonferroni's test or Student's t-test to determine differences in average scores according to the basic attributes. All statistical analyses were performed using IBM SPSS Ver. 25, with the significance level set at 5%.

Because descriptions varied from one nurse to another and some descriptions were not directly related to the issues and difficulties that nurses faced in providing support for sexual and reproductive problems among AYA, the first author needed to select meaningful descriptions at the start of the analysis. The first author then slightly edited each description without changing the meaning for analysis, as required. In the third step, authors bundled several descriptions of the same meaning into short sentences as codes. Once commonalities and differences in the codes were analyzed, several similar codes were categorized. Three researchers were involved in the coding and categorization process and performed their own analysis to verify the agreement rate (75.5%) and ensure validity. The codes that did not match among the three researchers were classified into the codes that were agreed upon after a discussion among the authors.

## Ethical considerations

Participants were debriefed in writing regarding the purpose of the survey, their freedom in research participation, the anonymity of the responses, and the absence of any consequences from opting out of the survey. Consent for participating in the study was obtained by nursing directors, and participants were encouraged to answer the questionnaire on their own volition.

The study protocol was approved by the research ethics review board of the Nagoya Medical Center Research Institute.

## Results

### Nurses' background

A total of 2,728 nurses participated in this study, and 1,982 responses were obtained from the nurses (recovery rate: 72.7%). Among the 2,728 nurses, 865 were considered effective respondents (31.7%).

Participants were affiliated with the following departments: 106 (12.3%) with "Pediatrics (Mixed)," including Pediatrics, Pediatric Surgery, Pediatric Hematology & Oncology, Infant/

Child Internal Medicine, and departments that treated both children and adults; 153 (17.7%) with "Gynecology/Breast Oncology"; and 606 (70.1%) with "Other Departments for Adults," including Surgery, Internal Medicine, Hematology & Oncology (Adults), and others. A total of 64 (7.4%) answered that they were certified or specialized nurses. Moreover, 421 participants (48.7%) had a nursing experience of >10 years, 353 (40.8%) had an experience of 4–9 years in managing patients with cancer, and 110 (12.7%) were affiliated with a cancer-related academic society (Table 1).

## Nurses' perceptions on support for sexual and reproductive issues among AYA with cancer

Questions regarding sexual and reproductive issues among AYA with cancer that were scored the highest included "Do you think that a sufficient support system for sexual and reproductive functions is in place?" (2.78 ± 0.76; range, 1–4), followed by "Do you think that adolescent patients with cancer (age, 15–19 years) are sufficiently debriefed with the necessary information regarding their sexual and reproductive functions?" (2.49 ± 0.74; range, 1–4) (Table 2). Cronbach's coefficient α for the subscale was 0.832.

A comparison of the scores according to participants' attributes showed that those who were affiliated with adult-related departments, who had extensive experience as a nurse or cancer nurse, who were affiliated with cancer-related academic and professional societies, and who were specialists/certified nurses had significantly higher scores (Fig 1).

## Challenges with supporting sexual and reproductive issues among aya with cancer

A total of 135 free descriptions were obtained from 128 participants who expressed their challenges and difficulties they faced while providing support for sexual and reproductive problems. After excluding 33 descriptions that omitted challenges or difficulties related to

**Table 1. Nurse's background.** n = 865.

|  | n (%) |
|---|---|
| Affiliated departments |  |
| Pediatrics(Mixed) | 106 (12.3) |
| Gynecology/Breast Oncology | 153 (17.7) |
| Other Departments for Adults | 606 (70.1) |
| Certified or specialized nurses |  |
| Yes | 64 (7.4) |
| No | 801 (92.6) |
| Total years for nursing |  |
| 1–3 | 165 (19.1) |
| 4–9 | 279 (32.3) |
| ≧10 | 421 (48.7) |
| Total years for oncology nursing |  |
| 1–3 | 227 (26.2) |
| 4–9 | 353 (40.8) |
| ≧10 | 285 (32.9) |
| Cancer-related academic society |  |
| Belonged | 110 (12.7) |
| Not belonged | 755 (87.3) |

**Table 2. Nurse's perception of support for sexual and reproductive issues among AYA.**  n = 865.

| | Sufficient n (%) | Moderately Sufficient n (%) | Moderately Insufficient n (%) | Insufficient n (%) | M(SD)* |
|---|---|---|---|---|---|
| Q4.Do you think that a sufficient support system for sexual and reproductive functions is in place? | 24 (2.8) | 287 (33.2) | 405 (46.8) | 149 (17.2) | 2.78 (0.76) |
| Q1.Do you think that adolescent patients with cancer are sufficiently debriefed with necessary information regarding their sexual and reproductive functions? | 52 (6.0) | 413 (47.7) | 326 (37.7) | 74 (8.6) | 2.49 (0.74) |
| Q3.Do you think that examinations and care are conducted sufficiently taking sexuality into consideration? | 46 (5.3) | 494 (57.1) | 275 (31.8) | 50 (5.8) | 2.38 (0.68) |
| Q2.Do you think that young adult patients with cancer are sufficiently debriefed with necessary information regarding their sexual and reproductive functions? | 152 (17.6) | 473 (54.7) | 202 (23.4) | 38 (4.4) | 2.15 (075.) |

*Mean score/ Sufficient(1point), Moderately Sufficient(2points), Moderately Insufficient(3points), Insufficient(4points)

sexuality, 103 descriptions were classified, of which 27 codes and 9 categories were extracted. The category, code, and description examples are shown in Tables 3 and 4.

Challenges experienced by healthcare providers attempting to provide support were classified into the following three categories: [Poor support by healthcare providers], which included the codes "Dealing with private matters" and "Lack of knowledge and experience"; [Insufficient support system], which included the codes "Insufficient system for providing

| | M (SD) | p |
|---|---|---|
| Affiliated departments [b] | | |
| Pediatrics (Mixed) | 9.51 (2.08) | |
| Gynecology/Breast Oncology | 9.29 (2.30) | ** b |
| Other Departments for Adults | 9.98 (2.43) | |
| Certified or specialized nurses | | |
| Yes | 10.73 (2.69) | ** a |
| No | 9.72 (2.34) | |
| Total years for nursing | | |
| 1-3 | 9.19 (1.99) | |
| 4-9 | 9.67 (2.28) | * / *** b |
| ≧10 | 10.12 (2.53) | |
| Total years for oncology nursing | | |
| 1-3 | 9.42 (2.12) | |
| 4-9 | 9.82 (2.37) | ** b |
| ≧10 | 10.08 (2.55) | |
| Cancer-related academic society | | |
| Belonged | 10.41 (2.32) | * a |
| Not belonged | 9.71 (2.70) | |

[a]t-test、 [b] Bonferroni's-test、      * $p < .05$      ** $p < .01$      ***$P < .001$

**Fig 1. Comparison of the scores on nurse's perception.**

**Table 3. Challenges related to the healthcare support system (free descriptions).** n = 53.

| Category | Code (n) | Description example |
|---|---|---|
| Poor support by healthcare providers | Dealing with private matters (10) | "It is difficult to know how far I should go to support sexual function." |
| | Lack of knowledge and experience (7) | "I think the awareness of nurses about fertility is low." |
| | Timing of discussion about sexual matters (6) | "It is difficult to talk without a good relationship." |
| | Emotional support (5) | "I am confused about how to deal with emotional depression." |
| | Building a relationship with young patients (4) | "It is difficult to build and maintain a confidential relationship with this generation." |
| Insufficient support system | Insufficient system for providing information (6) | "The explanation from doctors is insufficient, and sometimes patients cannot understand it." |
| | Closed culture regarding sex (4) | "In Japan, sex is considered to be taboo, and it is very difficult to talk about sex to patients during adolescence." |
| | Poor cooperation of the multidisciplinary team (3) | "I want to work with a specialist from a mental health center when I talk with teen patients." |
| | Insufficient economic support system (1) | "Egg freezing is expensive with long term costs, but there are few economic support systems." |
| Hesitant to intervene in sexual matters | Support for patients of the opposite sex (4) | "I hesitate to support patients of the opposite sex about sexual matters." |
| | Shame of unmarried and young nurses (2) | "It is difficult to explain because I do not have a partner and cannot understand the patients' problems." |
| | Hesitant to talk about sex with young patients (1) | "I do not know how far I should go because the patients are of the same generation as me." |

information" and "Closed culture regarding sex"; and [Hesitant to intervene in sexual matters], which included codes "Support for patients of the opposite sex."

Difficulties in providing support related to patient characteristics were classified into the following six categories. In accordance, [Respects the will of young patients] was most frequently reported by healthcare providers. This indicated that there were situations wherein family members had already made a decision before nurses could ask the opinion of the patient or a sufficient explanation was not provided to the adolescent at the request of the parents. In addition nurses experienced difficulties in [Situation where treatment is prioritized], including "Timing of fertility preservation." Even when nurses could provide explanation to the patients, several issues were noted, including "Postponed sexual and reproductive problems". Considering that AYA value their relationships with both partners and family members, adjustments are necessary when difference in opinions exists between patients and family members and between family members and partners. In such cases, nurses experienced difficulties in [Involvement with their families and partners]. Furthermore, AYA do not express themselves much, leading to frequent misinterpretations with respect to their feelings regarding sexual issues. As such, nurses responded that [Understanding the needs of young patients] was difficult and experienced difficulties in [Obtaining an understanding of disease]. Moreover, nurses noted difficulties in [Support for sexual or reproductive dysfunction] who experienced loss of fertility, who could not continue fertility treatment, and who had sexual dysfunction.

## Discussion

### Current trends in the support for issues involving sexual and reproductive functions and difficulties in providing care

The present study showed that nurses in the adult departments, certified nurse specialists/certified nurses, those with more experience in cancer nursing, and those affiliated with cancer-

**Table 4. Difficulties in providing support due to patient characteristics (free descriptions).** n = 50.

| Category | Code (n) | Description example |
|---|---|---|
| Respects the will of young patients | Family wishes are prioritized (7) | "When treatment starts immediately after diagnosis, sometimes family members make decisions without consulting the patient." |
| | Conflict between treatment and fertility (4) | "We think patients can have children if they live. However, it is difficult to support the decisions of patients who have conflicts about the possible loss of fertility." |
| | Decision-making for an uncertain future (2) | "I face difficulties in supporting decision-making because the thoughts of the patient may change in the future when they have a partner." |
| Situation where treatment is prioritized | Timing of fertility preservation (7) | "Sometimes treatment is postponed even if it should be started immediately because patients cannot give up egg retrieval." |
| | Postponed sexual and reproductive problems (5) | "Because the patients' top priority is life, they cannot think about sex or reproduction. In most cases, even if I explain it to them, they do not understand." |
| Involvement with their families and partners | Coordination of opinions between patients and family members (3) | "Nurses face difficulties when there is difference in opinions of patients and family members." |
| | Relationships between family members and partners (3) | "The key person may be the parents or a partner. In such cases, it is difficult to adjust when the relationship between the parents or a partner is complicated." |
| | Promoting understanding of family members or partners (2) | "It is difficult, even if I explain about sex, when the patients and their partners do not consider it important." |
| Understanding the needs of young patients | Patients who do not express themselves much (4) | "Young patients do not express themselves much, and it is difficult to know how much they understand." |
| | Understanding the real thoughts of patients (2) | "It is difficult to discuss what patients are really thinking about their problems regarding sex while maintaining privacy." |
| Obtaining an understanding of disease | Patients who have difficulties in understanding their disease condition (2) | "It is difficult to support patients who developed the disease at a young age and are always positive, but whose understanding of the disease is poor." |
| | Providing explanations for understanding (2) | "Many adolescent patients do not have clear thoughts about sex and it is difficult to judge whether they can understand the explanation." |
| | Patients who have difficulties accepting the diagnosis (1) | "It is difficult to support patients who cannot accept the disease because they are young." |
| Support for sexual or reproductive dysfunction | Providing emotional support for patients facing loss of fertility (5) | "It is difficult to support patients who want to have children but will lose fertility because of treatment." |
| | Consulting about sexual dysfunction (1) | "It is very difficult to support patients who cannot have a sex life following treatment." |

related academic and professional societies recognized that support was insufficient and faced challenges with supporting patients who had problems related to sexual and reproductive function. Our findings showed that nurses with specialized knowledge and advanced practices are more likely to recognize barriers and difficulties in supporting AYA with cancer. With the rapid development of reproductive medicine for cancer survivors, care providers and patients need to discuss reproductive concerns. The Japanese Clinical Practice Guidelines for Fertility Preservation in Pediatric, Adolescent, and Young Adults with Cancer specify that the oncologist should initiate the discussion on the possibility of infertility with patients receiving cancer treatment during their reproductive years. Furthermore, the oncologist should consider the availability and timing of fertility preservation options in close collaboration with specialists in reproductive medicine [13]. Therefore, in Japan, the task often falls on the oncologist to inform patients regarding possible infertility. A previous study reported that clinicians' unfamiliarity with infertility risks, fertility preservation technologies, referral processes, and procedures and environmental factors and their perceptions of fertility preservation influenced their practices regarding fertility discussions [14]. Participants were aware of the importance of sharing information; however, they may have felt that the explanation of the oncologist was insufficient. In addition, nurses who do not participate in the discussion may not understand the needs of AYA with cancer. Studies have demonstrated several challenges, including a lack of knowledge resulting in the avoidance of discussing fertility issues [15] and few opportunities for discussing fertility preservation immediately after diagnosis [16]. Moreover, a previous

report has shown that nurses were concerned about exacerbating negative emotions and the decision-making capacity of young patients [17]. To understand the needs of AYA with cancer and provide proper support and resources immediately after diagnosis, nurses caring for such patients on a daily basis should be more involved in discussions.

In the free description section of the questionnaire, nurses mentioned challenges with respect to the support system, including [Hesitant to intervene in sexual matters] and [Insufficient support system]. Nurses are hesitant to intervene in patient privacy and sexual and fertility issues because Japanese culture tends to view sexual issues as a taboo. Moreover, given that encounters with, and therefore intervention in, sexual issues due to cancer and cancer treatment among AYA are relatively rare for nurses, they consequently lack the professional knowledge and awareness to respond appropriately to the young patients' needs. Saunamaki et al [18] stated that although discussing sexual issues with patients is challenging for nurses, establishing an environment where nurses can discuss and understand these issues is an important approach toward providing support for their patients. Moreover, it is desirable to establish an environment where nurses can freely discuss issues with their professional team or ask where or with whom they can talk about problems related to sexual and reproductive function among AYA with cancer.

## Difficulty in providing support to AYA

Adolescent patients are expected to face difficulties while deciding about their future. The present study revealed that nurses recognized that providing information on sexual and reproductive problems to adolescent patients had a lesser impact than providing it to young adult patients because of psychosocial development. In Japanese culture, fertility is an important condition for marriage. Fertility may affect the adolescents' development of their sexual identity, particularly for those with fertility-related physical and functional health problems. Under such circumstances, some adolescent patients develop into adults with a vague sense of unease regarding future pregnancy and delivery or a feeling of loneliness in addition to a sense of withdrawal from talking about their problems with others [19].

Regarding the provision of support as per the characteristics of AYA, nurses faced challenges or difficulties in [Respect the will of young patients] and [Situation where treatment is prioritized]. Considering the improvements in cancer reproductive medicine, the possibility for retaining fertility among such patients has continued to improve. However, studies have suggested that the urgency to initiate treatment as well as the inadequacy of information have hindered patients from utilizing fertility preservation services [20, 21]. When providing information regarding fertility preservation treatment, nurses must convey the information and also support such patients in making decisions regarding their life after completion of treatment while facing the shock of a cancer diagnosis. Nurses need to support decision-making based on the opinions and future of the patients while adjusting for relationships among the patient, family members, and partner. Apart from cancer diagnosis and treatment, problems related to sexuality and reproduction affect the future of AYA. As such, nurses are required to deal with these difficult challenges while understanding patients' values and complying with their wishes. Our findings suggested that nurses were aware of the importance of supporting sexual and reproductive issues but faced difficulties in approaching them, particularly with irreversible issues. Similarly, in the services of adult cancer survivors, the smallest number of practitioners felt they had skills to manage sexual and reproductive issues. Nevertheless, training requirements were not highly prioritized [22]. Nurses may need continuous education to improve communication skills with the young generation and competently handle sexual and reproductive issues. Building multidisciplinary teams for supporting AYA regarding these

issues is necessary to manage impacts on this generation's sense of identity and ability to adopt a more positive life meaning.

## Limitations

Although the present study targeted nurses working in departments related to AYA with cancer and survivors, several missing values were noted in the survey responses, resulting in a low effective response rate. Nonetheless, the current study could elucidate the issues and difficulties the nurses face while supporting sexual and reproductive issues among AYA, suggesting the need for enhancing support and for considering policies that further nurses' expertise.

## Conclusions

The present study revealed that some nurses recognized the insufficiency of the current support system and the level of provided information regarding sexual and reproductive issues faced by AYA with cancer and survivors. Nurses who had extensive experience, who were affiliated with adult-related departments, and who were specialists/certified had a higher level of awareness. Moreover, our findings clarified the challenges and difficulties that nurses face regarding sexual and reproductive issues. Overall, the study suggests that the improvement of the healthcare system to enhance the quality of care provided specifically for AYA is necessary.

## Supporting information

**S1 File. Questionnaire (English).**
(PDF)

**S2 File. Questionnaire (Japanese).**
(PDF)

**S1 Data.**
(XLSX)

## Acknowledgments

We are deeply grateful to all the staff members who participated in this study's survey.

## Author Contributions

**Conceptualization:** Akiko Tomioka, Mitsue Maru.

**Formal analysis:** Akiko Tomioka, Kyoko Obama, Mitsue Maru.

**Funding acquisition:** Mitsue Maru.

**Investigation:** Akiko Tomioka, Kyoko Obama, Hiromi Okada, Eiko Yamauchi, Kimiko Iwase, Mitsue Maru.

**Methodology:** Mitsue Maru.

**Project administration:** Mitsue Maru.

**Validation:** Kyoko Obama, Hiromi Okada, Eiko Yamauchi, Kimiko Iwase, Mitsue Maru.

**Visualization:** Akiko Tomioka.

**Writing – original draft:** Akiko Tomioka.

**Writing – review & editing:** Kyoko Obama, Hiromi Okada, Eiko Yamauchi, Kimiko Iwase, Mitsue Maru.

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
