## [Decision Letter · Decision Letter 0]

26 Nov 2021

PONE-D-21-18301Nurse’s Perceptions of Support for Sexual and Reproductive Issues in Adolescents and Young Adults with CancerPLOS ONE

Dear Dr. 富岡,

Thank you for submitting your manuscript to PLOS ONE. After careful consideration, we feel that it has merit but does not fully meet PLOS ONE’s publication criteria as it currently stands. Therefore, we invite you to submit a revised version of the manuscript that addresses the points raised during the review process.

Please update references

Please comment on the limitations of your study regarding aspects highlighted by reviewer #1

Please submit the paper to professional English proofreading and revised sentences according to reviewer #2

We look forward to receiving your revised manuscript.

Kind regards,

Marta Panzeri, Ph.D.

Academic Editor

PLOS ONE

Journal Requirements:

2. Please include additional information regarding the survey or questionnaire used in the study and ensure that you have provided sufficient details that others could replicate the analyses. For instance, if you developed a questionnaire as part of this study and it is not under a copyright more restrictive than CC-BY, please include a copy, in both the original language and English, as Supporting Information. If the original language is written in non-Latin characters, for example Amharic, Chinese, or Korean, please use a file format that ensures these characters are visible.

3. Please state whether you validated the questionnaire prior to testing on study participants. Please provide details regarding the validation group within the methods section.

Reviewers' comments:

Reviewer's Responses to Questions

**Comments to the Author**

1. Is the manuscript technically sound, and do the data support the conclusions?

Reviewer #1: Yes

Reviewer #2: Yes

2. Has the statistical analysis been performed appropriately and rigorously? 

Reviewer #1: I Don't Know

Reviewer #2: Yes

3. Have the authors made all data underlying the findings in their manuscript fully available?

Reviewer #1: Yes

Reviewer #2: Yes

4. Is the manuscript presented in an intelligible fashion and written in standard English?

Reviewer #1: Yes

Reviewer #2: No

5. Review Comments to the Author

Reviewer #1: The subject of the article is interesting and quite innovative, it explores at the same time sexual, medical and reproductive issues together with cultural aspects.

Here are listed some aspects to ameliorate the manuscript:

- some references need to be updated; e.g., King L, Quinn GP, Vadaparampil ST, Gwede CK, Miree CA, Wilson C, et al. Oncology nurses’ perceptions of barriers to discussion of fertility preservation with patients with cancer. Clin J Oncol Nurs. 2008; 12(3): 467-476.

- line 68: “The survey period was from June to August 2016” – Authors can clarify/explicit the the reason why the decided to delay the publication.

- lines 197-198: “The low effective response rate in this study was a combination of missing answers and the considerable number of nurses who answered “I don’t know,” indicating that nurses themselves are not sufficiently aware of the current situation”. This statement sounds as an inference or a personal authors’ deduction and needs better explanation.

- line 70 “measurements”: nurses’ perception investigation could have been more specific, using one or more standardized questionnaire in association with the four questions.

- a comparison with other researches would have been appreciated, even if there are few studies on this topic. It could be interesting a comparison with nurses’ perception of support for sexual and reproductive issues in a different age, or for a different medical problem, or related to other form of support.

Reviewer #2: This study meets the criteria for publication as it presents the results of primary scientific research and the results have not been published elsewhere. The statistical analyses are appropriate and described in detail. All ethical standards are met. The study adheres to appropriate reporting guidelines and community standards for data availability.

This articles does have grammatical errors. In addition, statements at times are written in a very definitive manner, when not appropriate. The authors would benefit from having someone with a more nuanced understanding of the English language review and correct the paper. Please see 2 examples below.

“In addition, cancer treatments, such as chemotherapy and radiation, cause fertility and sexual dysfunction.” The more appropriate sentence would be “In addition, cancer treatments such as chemotherapy and radiation, may cause fertility and sexual dysfunction.” Not all chemotherapy and radiation treatments cause problems with fertility and sexual dysfunction.

Another example- “Receiving a diagnosis and the subsequent treatment of cancer during this life stage inevitably disrupts their outlook regarding a promising future.” This statement is very definitive and assumes the future is not promising. While getting diagnosed and treated for cancer will cause some disruption in the life of AYA’s, their future can still be promising and some patients even report psychological benefits from the cancer experience. Perhaps a preferable way to phrase the statement- “Receiving a diagnosis and the subsequent treatment of cancer during this life stage inevitably disrupts their life outlook.”

These are just 2 examples found throughout the article. This reviewer suggests having the authors review for grammatical errors and correct. In addition, this reviewer suggests reviewing whether definitive statements have research data to back up the conclusions stated.

The organization of the “free descriptions” was confusing as there are too many codes and categories. The authors may want to consider a more succinct way of organizing and reporting the qualitative information.

6. PLOS authors have the option to publish the peer review history of their article (what does this mean?). If published, this will include your full peer review and any attached files.

Reviewer #1: **Yes: **Valentina Cosmi

Reviewer #2: No

---

## [Author Response · Author response to Decision Letter 0]

20 Jan 2022

Thank you for the opportunity to revise our manuscript. Please see our responses below.

Required 

Editor

1. Comments to author

Response

We ensured that the style requirements were met and modified the filename.

2. Comments to author

Please include additional information regarding the survey or questionnaire used in the study and ensure that you have provided sufficient details that others could replicate the analyses. For instance, if you developed a questionnaire as part of this study and it is not under a copyright more restrictive than CC-BY, please include a copy, in both the original language and English, as Supporting Information. If the original language is written in non-Latin characters, for example Amharic, Chinese, or Korean, please use a file format that ensures these characters are visible.

Response

We have added Japanese and English questionnaires to Supporting Information.

3. Comments to author

Please state whether you validated the questionnaire prior to testing on study participants. Please provide details regarding the validation group within the methods section.

Response

We have stated that we have validated the questionnaire as follows:

“The researcher’s group consisting of four pediatric cancer nurses, two adult cancer nurses, and two clinical nurses reviewed the questionnaire for surface and content validity.”

4. Comments to author

In your Data Availability statement, you have not specified where the minimal data set underlying the results described in your manuscript can be found. PLOS defines a study's minimal data set as the underlying data used to reach the conclusions drawn in the manuscript and any additional data required to replicate the reported study findings in their entirety. All PLOS journals require that the minimal data set be made fully available. For more information about our data policy, please see http://journals.plos.org/plosone/s/data-availability

Response

We have added the dataset to Supporting Information.

5. Comments to author

Response

We have revised the reference list and provided a rebuttal letter.

Reviewer 

We wish to express our sincere appreciation to the reviewers for their insightful comments on our manuscript. As per the reviewers’ suggestions, we have revised the manuscript, and our responses to the reviewers’ comments are as follows:

Reviewer #1:

1. Comments to author

some references need to be updated; e.g., King L, Quinn GP, Vadaparampil ST, Gwede CK, Miree CA, Wilson C, et al. Oncology nurses’ perceptions of barriers to discussion of fertility preservation with patients with cancer. Clin J Oncol Nurs. 2008; 12(3): 467-476.

Response

Thank you for pointing this out. We have updated reference number14.

2. Comments to author

line 68: “The survey period was from June to August 2016” – Authors can　clarify/explicit the reason why the decided to delay the publication.

Response

This study was part of a large survey on physicians, nurses, and patients across Japan in 2016, and after analyzing several surveys, we found it useful to focus and report on nurse’s perception of support for sexual and reproductive issues. Therefore, we decided to submit this manuscript. We have stated that this survey was part of a large study as follows:

“This study was part of a large survey across Japan. The questionnaire had six sections: a) care difficulties for AYA cancer patients/survivors, b) perceived needs of patients/survivors in daily care, c) support for sexual and reproductive issues, d) end-of-life care, e) characteristics of AYA patients/survivors whom nurses felt care difficulties, and f) facilitating/disturbing factors for the quality of care for AYAs. In this study, we used the questionnaire section ‘support for sexual and reproductive issues’.”

3. Comments to author

lines 197-198: “The low effective response rate in this study was a combination of missing answers and the considerable number of nurses who answered “I don’t know,” indicating that nurses themselves are not sufficiently aware of the current situation”. This statement sounds as an inference or a personal authors’ deduction and needs better explanation.

Response

Thank you for this suggestion. We agree with the reviewer and have removed the sentence because the low effective response does not indicate that the nurses are not sufficiently aware of the current situation.

4. Comments to author

line 70 “measurements”: nurses’ perception investigation could have been more specific, using one or more standardized questionnaire in association with the four questions.

Response

Thank you for this suggestion. This was the first exploratory study involving a multidisciplinary team in Japan, and we focused on the background and current status of the subjects. The survey questionnaire consisted of six sections and had many questions, so we did not use the standardized questionnaire for practical reasons. We would like challenge this with further research.

5. Comments to author

a comparison with other researches would have been appreciated, even if there are few studies on this topic. It could be interesting a comparison with nurses’ perception of support for sexual and reproductive issues in a different age, or for a different medical problem, or related to other form of support.

Response

Thank you for this suggestion. We have added the following text to the Discussion section to compare this study to previous studies investigating the perceptions of healthcare providers in the long-term care of adult cancer survivors:

“Similarly, in the services of adult cancer survivors, the smallest number of practitioners felt they had skills to manage sexual and reproductive issues. Nevertheless, training requirements were not highly prioritized［22］.”

Reviewer #2: 

1. Comments to author

This articles does have grammatical errors. In addition, statements at times are written in a very definitive manner, when not appropriate. The authors would benefit from having someone with a more nuanced understanding of the English language review and correct the paper. Please see 2 examples below.

“In addition, cancer treatments, such as chemotherapy and radiation, cause fertility and sexual dysfunction.” The more appropriate sentence would be “In addition, cancer treatments such as chemotherapy and radiation, may cause fertility and sexual dysfunction.” Not all chemotherapy and radiation treatments cause problems with fertility and sexual dysfunction.

Another example- “Receiving a diagnosis and the subsequent treatment of cancer during this life stage inevitably disrupts their outlook regarding a promising future.” This statement is very definitive and assumes the future is not promising. While getting diagnosed and treated for cancer will cause some disruption in the life of AYA’s, their future can still be promising and some patients even report psychological benefits from the cancer experience. Perhaps a preferable way to phrase the statement- “Receiving a diagnosis and the subsequent treatment of cancer during this life stage inevitably disrupts their life outlook.”

These are just 2 examples found throughout the article. This reviewer suggests having the authors review for grammatical errors and correct. In addition, this reviewer suggests reviewing whether definitive statements have research data to back up the conclusions stated.

Response

 Thank you for this suggestion. We have revised two sentences based on your suggestion. Moreover, we have proofread the manuscript again.

　

2. Comments to author

The organization of the “free descriptions” was confusing as there are too many codes and categories. The authors may want to consider a more succinct way of organizing and reporting the qualitative information.

Response

Thank you for this suggestion. We have simplified the code because there are several qualitative codes. Moreover, we have shown only one.

All abovementioned changes have been reflected in the manuscript in red text.

We hope you find the revised manuscript acceptable for publication. Thank you once again for the consideration.

---

## [Decision Letter · Decision Letter 1]

9 Mar 2022

Nurse’s Perceptions of Support for Sexual and Reproductive Issues in Adolescents and Young Adults with Cancer

PONE-D-21-18301R1

Dear Dr. 富岡,

We’re pleased to inform you that your manuscript has been judged scientifically suitable for publication and will be formally accepted for publication once it meets all outstanding technical requirements.

Kind regards,

Marta Panzeri, Ph.D.

Academic Editor

PLOS ONE

Additional Editor Comments (optional):

Reviewers' comments:

Reviewer's Responses to Questions

**Comments to the Author**

1. If the authors have adequately addressed your comments raised in a previous round of review and you feel that this manuscript is now acceptable for publication, you may indicate that here to bypass the “Comments to the Author” section, enter your conflict of interest statement in the “Confidential to Editor” section, and submit your "Accept" recommendation.

Reviewer #1: All comments have been addressed

Reviewer #2: All comments have been addressed

2. Is the manuscript technically sound, and do the data support the conclusions?

Reviewer #1: (No Response)

Reviewer #2: Yes

3. Has the statistical analysis been performed appropriately and rigorously? 

Reviewer #1: (No Response)

Reviewer #2: Yes

4. Have the authors made all data underlying the findings in their manuscript fully available?

Reviewer #1: (No Response)

Reviewer #2: Yes

5. Is the manuscript presented in an intelligible fashion and written in standard English?

Reviewer #1: (No Response)

Reviewer #2: Yes

6. Review Comments to the Author

Reviewer #1: (No Response)

Reviewer #2: The suggested changes were made. The only additional change relates to the first sentence in the conclusions section. Please change the word "several" to "some."

7. PLOS authors have the option to publish the peer review history of their article (what does this mean?). If published, this will include your full peer review and any attached files.

Reviewer #1: **Yes: **Valentina Cosmi

Reviewer #2: No

---

## [Editor Report · Acceptance letter]

14 Mar 2022

PONE-D-21-18301R1 

Nurse’s perceptions of support for sexual and reproductive issues in adolescents and young adults with cancer 

Dear Dr. Tomioka:

I'm pleased to inform you that your manuscript has been deemed suitable for publication in PLOS ONE. Congratulations! Your manuscript is now with our production department. 

Kind regards, 

on behalf of

Dr. Marta Panzeri 

Academic Editor

PLOS ONE